

# The prognostic impact of age in different molecular subtypes of breast cancer: a population-based study

Dongjun Dai[1,*], Yiming Zhong[1,*], Zhuo Wang[1], Neelum Aziz Yousafzai[1], Hongchuan Jin[2] and Xian Wang[1]

[1] Department of Medical Oncology, Sir Run Run Shaw Hospital, Medical School, Zhejiang University, Hangzhou, Zhejiang, China
[2] Laboratory of Cancer Biology, Key Lab of Biotherapy, Sir Run Run Shaw Hospital, Medical School, Zhejiang University, Hangzhou, Zhejiang, China
[*] These authors contributed equally to this work.

## ABSTRACT

**Background**. The aim of current study was to use competing risk model to calculate the potential differences that age played in the prognosis of different breast cancer subtypes.
**Methods**. The cohort was selected from Surveillance, Epidemiology, and End Results (SEER) program. The cumulative incidences of death (CID) was assessed for breast cancer caused deaths and other causes of mortality. The multivariate Cox proportional hazards regression model and the multivariate subdistribution hazard (SH) model were used to evaluate the prognostic value of age in different breast cancer subtypes.
**Results**. We involved 33,968 breast cancer patients into our cohort. We found older patients had worse overall survival (OS) than young patients in hormone receptor positive and human epidermal growth factor receptor 2 positive breast cancer (HR+/HER2+) ($\geq$40 vs. <40, HR = 2.07, 95% CI [1.28–3.35], $p < 0.05$). However, when we used competing risk model, we found young age was an independent risk factor only for triple negative breast cancer (TNBC) ($\geq$40 vs. <40, HR = 0.71, 95% CI [0.56–0.89], $p < 0.05$). No association was found in other groups.
**Conclusion**. Our research was currently the largest sample size study and the first competing risk model-based study on the prognostic association between age and different breast cancer subtypes. We found <40 years patients had worse breast cancer specific survival (BCSS) than older patients in the TNBC subtype.

# INTRODUCTION

Breast cancer is a heterogeneous disease. Subtypes of breast cancer show diverse phenotype and have different responsiveness to treatments (*Perou et al., 2000*). There are four main molecular subtypes of breast cancer that comprise luminal A, luminal B breast cancer, basal-like and HER2-like breast cancer. All these types of breast cancer were determined by expression of ER (estrogen receptor), PR (progesterone receptor), HER2 (human epidermal growth factor receptor 2) and proliferative markers such as Ki67 and cytokeratin

Corresponding author
Xian Wang, wangx118@zju.edu.cn

CK5/6 (*Reis-Filho & Pusztai, 2011*). Triple-negative breast cancer (TNBC) is referred to breast cancer with negative expression of ER, PR and HER2. Most of TNBC are genetically defined basal subtype (*Alluri & Newman, 2014*). The American Cancer Society showed that in different races, the incidence rates per 100,000 of luminal A, luminal B, HER2 and TNBC subtypes range from 53–82, 11–14, 4–7 and 8–24, respectively.

Different breast cancer subtypes have different prognostic and therapeutic implications (*Haque et al., 2012*; *Hennigs et al., 2016*; *Sorlie et al., 2001*). Luminal A patients often have low-grade tumors, and good prognosis (*Blows et al., 2010*; *Perou & Borresen-Dale, 2011*). Luminal B, HER2 and TNBC types are widely recognized to have poorer survival and tumors with higher grade (*Tran & Bedard, 2011*). Hormone therapy is used to treat hormone receptors positive (HR+) breast cancer (Luminal A and Luminal B) (*Abdulkareem & Zurmi, 2012*). HER2 inhibitor, such as trastuzumab, is usually administrated to HER2 and Luminal B breast cancers (*Kim et al., 2017*). For TNBC patients, chemotherapy remains the mainstay of treatment (*Lebert et al., 2018*).

Age at diagnosis of the patient was found to be an important prognostic factor for breast cancer (*Beadle, Woodward & Buchholz, 2011*). Young age at diagnosis was observed to be correlated with worse prognosis (*Anders et al., 2008*; *Anderson et al., 2009*). Breast cancers in younger women were found to have lower mRNA expression of ER and PR, and higher expression of HER2 and EGFR (*Anders et al., 2008*). Previous study showed breast cancer in young patients was more aggressive (ER-negative and basal-like subtype) and it was more indolent (ER-positive and non-basal-like subtype) in older female breast cancer patients (*Anderson et al., 2014*). Whether different age frequencies could alter the prognosis in different breast cancer subtypes was currently a subject of discussion or speculation. Previous study found that in hormone receptor (HR) positive patients (ER or PR positive), the younger ones (<40 years) had a worse relapse-free-survival (RFS) than older ones. While there was no association between age and RFS in HER2 or TNBC subtypes (*Azim Jr et al., 2012*; *Liu et al., 2014*). On the contrary, other studies had found <40 years patients had a worse overall survival (OS) than older ones in TNBC subtype but not the HR+ or HER2 subtypes (*Liedtke et al., 2013*; *Liedtke et al., 2015*).

It should be noted that breast cancers have a relative long survival rate, which means that the OS might be influenced by other causes of death. A competing risk model considers both the disease-specific death and other causes of death, which is widely used for prognostic analysis of long survival disease (*Latouche et al., 2013*; *Satagopan et al., 2004*). There was no study using the competing risk model to calculate the prognostic value of age in different breast cancer molecular subtypes. Besides, previous studies investigated with relative small-scale samples. The Surveillance, Epidemiology, and End Results (SEER) database of the National Cancer Institute is a national collaboration program of the United State, covering almost 26% of the population of the United States for cancer incidence and survival data. To our best knowledge, there was no SEER-based study on the association between age and the prognosis in breast cancer subtypes. Hence, the current study performed a competing risk model-based analysis of age in the prognosis of breast cancer subtypes using SEER-based population.

## MATERIAL AND METHODS

### Patient screening

We selected our cohort from SEER by using SEER*Stat 8.3.5 software (SEER ID: daid). Since the five-year relative survival rates of breast cancer was over 90%, long-term follow-up of breast cancer patients were collected. Besides, as HER2 status was registered since 2010, we involved it with the patients diagnosed in 2010. We used the follow inclusion criteria to screen the patients: (1) female primary breast cancer patients diagnosed between age 20 to79 who had done surgery; (2) records of ER, PR and HER2 status; (3) unilateral invasive ductal carcinoma with specific location; (4) include clinicopathological information of race, laterality, tumor location, grade, tumor size, 7th AJCC tumor stage, number of positive regional nodes, marital status, and records that whether the patients had radiotherapy or chemotherapy; (5) the survival time of the patient should be over 3 months, and the survival status should be recoded for survival analyses. Patients who did not meet these criteria would be excluded. It should be noted that the SEER database defined breast cancer subtypes by immunohistochemistry HR and HER2 status. Hence, the molecular subtypes of breast cancer in current study were roughly defined as the follows: HR+/HER2-, HR+/HER2+, HER2 and TNBC.

### Study variables and endpoints

We extracted the following variables from the selected cohorts that included age at diagnosis (20–39, 40–49, 50–79), race (Caucasian, African American, American Indian/Alaska Native, Asian or Pacific Islander), laterality (right or left side), tumor location, grade (well-differentiated, moderately differentiated, poorly differentiated, undifferentiated or anaplastic), tumor size, 7th AJCC tumor stage, number of positive regional nodes, marital status, and radiotherapy or chemotherapy experience. The tumor location was defined by SEER Site Specific Coding Modules (https://seer.cancer.gov/manuals/2016/appendixc.html), which comprised nipple, central portion of breast, upper-inner quadrant of breast, lower-inner quadrant of breast, upper-outer quadrant of breast, lower-outer quadrant of breast, axillary tail of breast, overlapping lesion of breast and breast that is not otherwise specified. The 7th AJCC tumor stage was roughly considered as I, II, III and IV. The widowed or single (never married or having a domestic partner) or divorced or separated patients was classified as unmarried. The value of tumor size and number of positive regional nodes were transformed into small categorical variables to fit the linear assumption. The median follow-up was estimated as the median observed survival time.

### Statistical analyses

The cumulative incidence of death (CID) was assessed for deaths caused by breast cancer and deaths from other causes. A multivariate SH model and a multivariate Cox proportional hazards regression model were used to assess the breast cancer specific survival (BCSS) and OS, respectively. The hazard ratio (HR) and 95% confidence index (95% CI) were calculated. All the statistical analyses were performed by using R version 3.4.4 (*R Core Team, 2018*). A *p*-value less than 0.05 was considered statistically significant.
## RESULTS

### Cohort selection

After selection, we involved 33,968 breast cancer patients into our cohort. The distribution of breast cancer molecular subtypes and age is listed in Table 1. The age <40 years presented with lower HR+ (72.67% vs. 82.28% and 84.61%, $p < 0.001$), more TNBC (20.14% vs. 13.32% and 11.28%, $p < 0.001$) and the HER2 subgroup (7.19% vs. 4.39% and 4.12%, $p < 0.001$). Besides, it was worth to notice that the age <40 years presented with more African Americans (15.13% vs. 11.86% and 9.44%, $p < 0.001$), higher grade (II–IV, 91.85% vs. 81.77% and 74.65%, $p < 0.001$), larger tumor size (>2 cm, 55.78% vs. 43.58% and 33.93%, $p < 0.001$), higher proportion of 7th AJCC tumor stage (II–IV, 69.53% vs. 55.48% and 43.84%, $p < 0.001$), more positive regional nodes (node >0, 46.56% vs. 37.87% and 29.33%, $p < 0.001$), less experience of radiotherapy (50.03% vs. 54.53% and 58.62%, $p < 0.001$) and more treatment of chemotherapy (80.50% vs. 63.16% and 39.22%, $p < 0.001$). The median follow-up time was 64 months. The other causes of mortality increased with the age of patients. The death rate caused by breast cancer and other cause were 11.03% and 1.60% for the age <40 years group, 5.74% and 1.88% for the age between 40 and 49 years group, and 5.82% and 5.65% for the age over 50 years group.

### The CIF analysis of age in breast cancer subtype prognosis

As shown in Fig. 1, competing risk model was used to evaluate the CID induced by age in different breast cancer subtypes. We found significant association between age and breast cancer subtypes in either the breast cancer caused deaths or other causes of death in the overall population and HR+ group ($p < 0.001$). The age <40 years group showed a worse outcome than older age groups. The CID caused by cancer was lower in HR+ group, which might be a result that HR+ subtype had a better prognosis than HER2 or TNBC subtypes. Further subgroup analysis found age <40 years could significantly increase the CID of HR+/HER2- ($p < 0.001$) and TNBC ($p < 0.001$). However, there was no association between age and the CID of HR+/HER2+ and HER2 group in our cohort ($p > 0.05$). As expected, the increased age could significantly elevate the CID of other causes of death in all groups ($p < 0.001$).

### Multivariate analysis of age and the prognosis of breast cancer specific survival (BCSS) and OS in different breast cancer subtypes

Next, we evaluated the independent prognostic value of age in breast cancer subtypes by performing multivariate Cox proportional hazards regression model and multivariate SH model, which included all variables we had obtained. As shown in Table 2, the multivariate Cox model showed that the older patients had worse OS than young patients in HR+/HER2+ subtype (≥40 vs. <40, HR = 2.07, 95% CI [1.28–3.35], $p < 0.05$), and there was no association between age and OS in other subtypes ($p > 0.05$). However, the SH model found the older patients did not show a worse prognosis in HR+/HER2+ subtype ($p > 0.05$), while the young patients were found to have a worse BCSS in TNBC (≥40 vs. <40, HR = 0.71, 95% CI [0.56–0.89], $p < 0.05$), which was conflict to the multivariate Cox regression analysis. Judged from CID plot (Fig. 1) of HR+/HER2+ subtype, the CID
**Table 1** The characteristic of each involved variables.

| Characteristics | <40y No. (%) | 40y–49y No. (%) | >50y No. (%) | *p* value |
|---|---|---|---|---|
| Breast cancer subtype | | | | <0.001 |
| HR+/HER2- | 1,023 (54.50%) | 4,506 (69.48%) | 19,351 (75.57%) | |
| HR-/HER2+ | 135 (7.19%) | 285 (4.39%) | 1,054 (4.12%) | |
| HR+/HER2+ | 341 (18.17%) | 830 (12.80%) | 2,313 (9.03%) | |
| HR-/HER2- | 378 (20.14%) | 864 (13.32%) | 2,888 (11.28%) | |
| Race | | | | <0.001 |
| White | 1,347 (71.76%) | 4,990 (76.95%) | 21,060 (82.25%) | |
| Black | 284 (15.13%) | 769 (11.86%) | 2,416 (9.44%) | |
| American Indian/Alaska Native | 13 (0.69%) | 44 (0.68%) | 113 (0.44%) | |
| Asian or Pacific Islander | 233 (12.41%) | 682 (10.52%) | 2,017 (7.88%) | |
| Laterality | | | | 0.02 |
| Right—origin of primary | 935 (49.81%) | 3,274 (50.49%) | 12,465 (48.68%) | |
| Left—origin of primary | 942 (50.19%) | 3,211 (49.51%) | 13,141 (51.32%) | |
| Location | | | | <0.001 |
| Nipple | 4 (0.21%) | 11 (0.17%) | 89 (0.35%) | |
| Central portion | 80 (4.26%) | 276 (4.26%) | 1,322 (5.16%) | |
| Upper-inner quadrant | 211 (11.24%) | 809 (12.47%) | 3,308 (12.92%) | |
| Lower-inner quadrant | 103 (5.49%) | 357 (5.51%) | 1,571 (6.14%) | |
| Upper-outer quadrant | 653 (34.79%) | 2,325 (35.85%) | 8,972 (35.04%) | |
| Lower-outer quadrant | 170 (9.06%) | 488 (7.53%) | 1,938 (7.57%) | |
| Axillary tail | 12 (0.64%) | 36 (0.56%) | 118 (0.46%) | |
| Overlapping lesion | 413 (22.00%) | 1,390 (21.43%) | 5,588 (21.82%) | |
| Breast, NOS | 231 (12.31%) | 793 (12.23%) | 2,700 (10.54%) | |
| Grade | | | | <0.001 |
| Well differentiated; Grade I | 153 (8.15%) | 1,182 (18.23%) | 6,492 (25.35%) | |
| Moderately differentiated; Grade II | 641 (34.15%) | 2,734 (42.16%) | 11,320 (44.21%) | |
| Poorly differentiated; Grade III | 1,074 (57.22%) | 2,540 (39.17%) | 7,701 (30.07%) | |
| Undifferentiated; anaplastic; Grade IV | 9 (0.48%) | 29 (0.45%) | 93 (0.36%) | |
| Tumor size | | | | <0.001 |
| ≤1 cm | 234 (12.47%) | 1,336 (20.60%) | 7,199 (28.11%) | |
| ≤2 cm | 596 (31.75%) | 2,323 (35.82%) | 9,719 (37.96%) | |
| ≤3 cm | 469 (24.99%) | 1,483 (22.87%) | 4,760 (18.59%) | |
| ≤4 cm | 267 (14.22%) | 611 (9.42%) | 1,794 (7.01%) | |
| ≤5 cm | 115 (6.13%) | 281 (4.33%) | 875 (3.42%) | |
| >5 cm | 196 (10.44%) | 451 (6.95%) | 1,259 (4.92%) | |
| Tumor stage | | | | <0.001 |
| I | 572 (30.47%) | 2,887 (44.52%) | 14,381 (56.16%) | |
| II | 897 (47.79%) | 2,572 (39.66%) | 8,132 (31.76%) | |
| III | 366 (19.50%) | 928 (14.31%) | 2,730 (10.66%) | |
| IV | 42 (2.24%) | 98 (1.51%) | 363 (1.42%) | |

**Table 1** (*continued*)

| Characteristics | <40y No. (%) | 40y–49y No. (%) | >50y No. (%) | *p* value |
|---|---|---|---|---|
| Regional nodes positive | | | | <0.001 |
| ≥10 | 87 (4.64%) | 211 (3.25%) | 808 (3.16%) | |
| 0 | 1,003 (53.44%) | 4,029 (62.13%) | 18,097 (70.67%) | |
| 1–3 | 616 (32.82%) | 1,735 (26.75%) | 5,196 (20.29%) | |
| 4–9 | 171 (9.11%) | 510 (7.86%) | 1,505 (5.88%) | |
| Marital status | | | | <0.001 |
| Married | 1,145 (61.00%) | 4,364 (67.29%) | 15,458 (60.37%) | |
| Unmarried | 732 (39.00%) | 2,121 (32.71%) | 10,148 (39.63%) | |
| Radiotherapy | | | | <0.001 |
| No | 938 (49.97%) | 2,949 (45.47%) | 10,596 (41.38%) | |
| Yes | 939 (50.03%) | 3,536 (54.53%) | 15,010 (58.62%) | |
| Chemotherapy | | | | <0.001 |
| No | 366 (19.50%) | 2,389 (36.84%) | 15,564 (60.78%) | |
| Yes | 1,511 (80.50%) | 4,096 (63.16%) | 10,042 (39.22%) | |

**Notes.**

Abbreviations: HR, hazard ratio; 95% CI, 95% confidence index; HR+, hormone receptor positive; HER2, human epidermal growth factor receptor 2; TNBC, triple negative breast cancer.

of breast cancer specific mortality (BCSM) of <40 years group had a similar tendency with CID of other causes of deaths of >50 years group, indicating that the worse OS of HR+/HER2+ in older patients might be a result of other causes of deaths. While as shown in CID plot of TNBC, the CID of BCSM of <40 years group had a much higher CID than other causes of deaths of >50 years group, therefore, no association of age and TNBC OS might be caused by other reasons of death.

Consistent with multivariate Cox model, our SH model found no association between age and HER2 subtype of breast cancer ($p > 0.05$).

### The association between treatment and prognosis of different breast cancer subtypes for <40 years patients

We further evaluated the association between treatment and prognosis of different breast cancer subtypes for <40 years patients. As shown in Table 3, the radiotherapy could improve the OS and CSS in HR+/HER2- group according to Cox and SH models, respectively. Chemotherapy could also increase the OS of HR+/HER2- breast cancer patients. The radiotherapy was also found to improve the OS in the HER2 group. However, this needed to be further validated because of the limited sample size of the HER2 group (radiotherapy vs. no radiotherapy, HR = 0.02, 95% CI [0.005–0.63], $p < 0.05$). It should be noted that there was no association between the radiotherapy and TNBC prognosis and, unexpectedly, the chemotherapy would worsen the CSS of TNBC patients in the SH model (chemotherapy vs. no chemotherapy, HR = 6.90, 95% CI [2.07–22.96], $p < 0.05$).

### DISCUSSION

Our study was currently the largest competing risk model-based study on the association between age and the prognosis in different breast cancer subtype. We observed <40 years

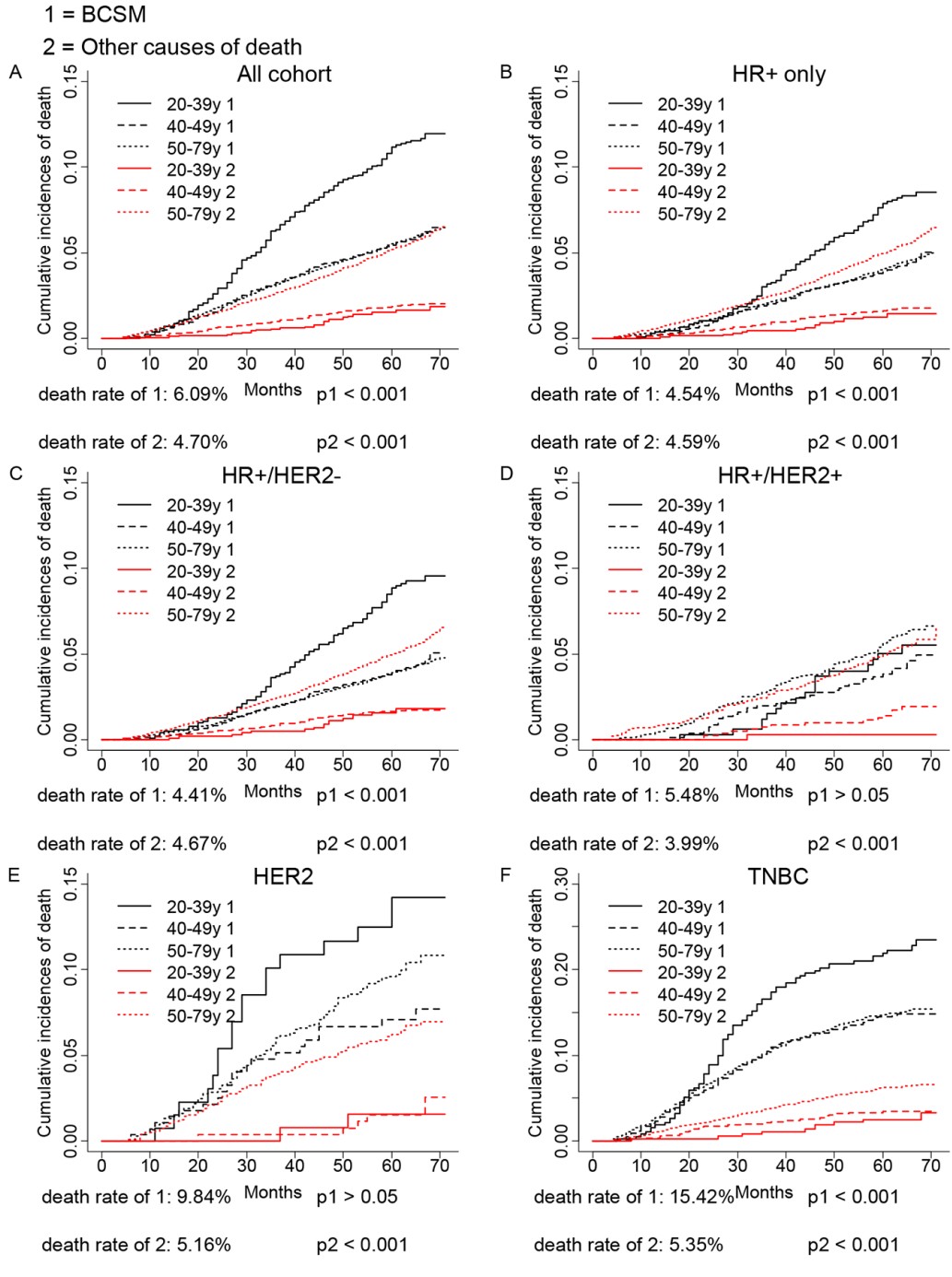

**Figure 1 Univariate analysis based on the competing risk regression model.** The association between age and breast cancer in all cohort (A), HR+ group (D), and molecular subtypes (C–F).

patients had a significant worse CID than older patients in HR+/HER2- and TNBC subtypes but not HR+/HER2+ and HER2 subtypes in breast cancer. Further Cox multivariate analyses found ≥40 years patients had a worse OS than <40 years patients in HR+/HER2+

**Table 2  Multivariate Cox and SH analyses breast cancer subtypes.**

|  | Age | OS<br>HR (95% CI) | SH<br>HR (95% CI) |
|---|---|---|---|
| All cohort | <40y | References | References |
|  | ≥40y | 1.12 (0.98–1.28) | 0.87 (0.75–1.02) |
| HR+ only | <40y | References | References |
|  | ≥40y | 1.27 (1.05–1.52) | 0.95 (0.77–1.17) |
| HR+/HER2- | <40y | References | References |
|  | ≥40y | 1.10 (0.91–1.35) | 0.86 (0.68–1.07) |
| HR+/HER2+ | <40y | References | References |
|  | ≥40y | **2.07 (1.28–3.35)** | 1.31 (0.78–2.20) |
| HER2 | <40y | References | References |
|  | ≥40y | 1.17 (0.73–1.88) | 0.86 (0.51–1.46) |
| TNBC | <40y | References | References |
|  | ≥40y | 0.87 (0.70–1.08) | **0.71 (0.56–0.89)** |

**Notes.**

Abbreviations: HR, hazard ratio; 95% CI, 95% confidence index; HR+, hormone receptor positive; HER2, human epidermal growth factor receptor 2; TNBC, triple negative breast cancer.

Significant results with $p < 0.05$ were bolded.

subtypes. However, this was eliminated when we applied the SH model, indicating that the worse OS in older patients was from other causes of death. Indeed, we observed the deaths rates caused by other reasons in >50 years patients was three times more than the <50 years patients. In addition, our SH model found <40 years patients had a worse BCSS than older patients in TNBC subtypes. Moreover, we found chemotherapy would worsen the CSS of TNBC patients under 40 years in SH model.

The previous study found <40 years patients presented with more African Americans, higher grade, higher tumor stage, more positive lymph nodes than older ones (*Anders et al., 2008*; *Anders et al., 2009*; *Bharat et al., 2009*), and young age patients presented with more aggressive subtypes such as HER2 and TNBC (*Anders et al., 2011*; *Anders et al., 2008*; *Anderson et al., 2014*; *Azim Jr et al., 2012*). Our study confirmed these clinicopathologic results. The menopause transition was found to influence the ER positive rate (*Tarone & Chu, 2002*; *Yasui & Potter, 1999*). Age-specific rates of ER- breast cancer cease to increase after 50 years of age, while the age-specific rates of ER+ breast cancer continue to increase after 50 years of age (*Tarone & Chu, 2002*), which might be explained by the finding that proliferation rate of ER+ cells increased with age (*Shoker et al., 1999*).

It is widely recognized that young breast cancer patients had worse outcomes than older ones (*Anders et al., 2009*; *Nixon et al., 1994*). The young people might not pay enough attention for breast cancer, which might lead to delay in diagnosis until worse stage has come. However, multivariate analysis found age was an independent factor associated with breast cancer (*Nixon et al., 1994*). Besides, younger women diagnosed with early-stage breast cancer were actually suggested to be more likely to die than older early-stage breast cancer patients (*Gnerlich et al., 2009*). The biological differences may distinct between the breast tumor of young and older patients. It was suggested that the incidence of breast cancer could be increased shortly after the first pregnancy (*Albrektsen et al., 2005*). The

**Table 3** Multivariate Cox and SH analyses of breast cancer subtypes.

| Group (number) | Treatment | <40y | |
|---|---|---|---|
| | | OS<br>HR (95% CI) | SH<br>HR (95% CI) |
| HR+/HER2- (1,023) | Radiotherapy | References<br>**0.60 (0.41–0.88)** | References<br>**0.59 (0.35–0.99)** |
| | Chemotherapy | References<br>**0.60 (0.38–0.96)** | References<br>0.69 (0.37–1.27) |
| HR+/HER2+ (341) | Radiotherapy | References<br>1.35 (0.53–3.45) | References<br>0.93 (0.29–3.01) |
| | Chemotherapy | References<br>0.39 (0.11–1.29) | References<br>0.93 (0.24–3.52) |
| HER2 (135) | Radiotherapy | References<br>**0.02 (0.005–0.63)** | References<br>0.53 (0.10–2.89) |
| | Chemotherapy | References<br>0.21 (0.005–9.07) | References<br>2.00 (0.29–13.80) |
| TNBC (378) | Radiotherapy | References<br>0.98 (0.51–1.88) | References<br>0.55 (0.23–1.35) |
| | Chemotherapy | References<br>1.17 (0.53–2.55) | References<br>**6.90 (2.07–22.96)** |

**Notes.**

HR, hazard ratio; 95% CI, 95% confidence index; HR+, hormone receptor positive; HER2, human epidermal growth factor receptor 2; TNBC, triple negative breast cancer.

Significant results with $p < 0.05$ were bolded. It should be noted that all the variables were involved in the multivariate Cox analyses. While in the Competing Risks Regression, in case the iterative algorithm was not converged, the race, laterality, tumor stage, marital status, positive regional nodes, radiotherapy and chemotherapy were involved in the SH model for HR+/HER2+ and the TNBC groups, the tumor stage, radiotherapy and chemotherapy were involved in SH model for HER2 group, and all variables were involved in SH model for the HR+/HER2- group.

pregnancy could increase the incidence of aggressive ER/PR (-) breast cancer while it decreased the incidence of ER/PR (+) tumors (*Britt, Ashworth & Smalley, 2007*; *Hildreth et al., 1983*; *Ruder et al., 1989*). Moreover, it was demonstrated that the gene expression pattern in breast cancers detected following a pregnancy was mainly attributable to TNBC, which was more prevalent in pregnancy-associated breast cancers than nulliparous group (*Asztalos et al., 2015*). We speculate that the gene pattern in TNBC is a key factor that related to its young age prevalence, such as BRAC1/2 mutation, which frequently occurred in both the young patients and TNBC patients (*Peshkin, Alabek & Isaacs, 2010*; *Rosenberg et al., 2016*). However, more gene pattern studies are required to get new evidences. In our study, competing risk model found that the age was an independent risk factor for TNBC but not the other molecular subtypes of breast cancer, which was consistent with previous studies (*Liedtke et al., 2013*; *Liedtke et al., 2015*).

Furthermore, in patients under 40 years, we found chemotherapy would worse the CSS of TNBC patients and have no association with the OS of TNBC patients. In contrast, chemotherapy would improve the OS of HR+/HER2- patients, and radiotherapy could improve the OS and CSS of HR+/HER2- patients. The different response in the treatment of different breast cancer subtypes might partially explain the worse prognosis of <40 years

TNBC patients and prove new hints for clinicians. However, the sample size of the <40 years patients in current study was relatively small. Further larger scale studies are needed for a more reliable result.

There are some limitations in our study. First, this is a retrospective study, which presented a relatively low level of clinical evidence. Further randomized controlled trials are required. Second, selection bias might exist as we only included patients with complete information for involved variables. Third, breast cancer subtypes are only defined by ER, PR and HER2 status in the SEER database; detailed molecular information such as Ki-67 and other proliferating factors (without which the luminal A and luminal B subtypes could not be distinguished) was missing. Further detailed studies with more specific molecular classification are required.

## CONCLUSIONS

In summary, the current study was the first competing risk model-based study with the largest sample size on the prognostic association between age and breast cancer subtypes. We found age <40 years was an independent risk factor for TNBC but not for the other subtypes of breast cancers.

### Funding

This grant was supported by the National Natural Science Foundation of China (81372178; 81502386; 81772944) and the Zhejiang Provincial Program for High-level Innovative Healthcare talents. The funders had no role in study design, data collection and analysis, decision to publish, or preparation of the manuscript.

### Grant Disclosures

The following grant information was disclosed by the authors:
National Natural Science Foundation of China: 81372178, 81502386, 81772944.
Zhejiang Provincial Program for High-level Innovative Healthcare talents.

### Competing Interests

The authors declare there are no competing interests.

### Author Contributions

- Dongjun Dai conceived and designed the experiments, prepared figures and/or tables, authored or reviewed drafts of the paper.
- Yiming Zhong and Neelum Aziz Yousafzai authored or reviewed drafts of the paper.
- Zhuo Wang analyzed the data, contributed reagents/materials/analysis tools.
- Hongchuan Jin analyzed the data, contributed reagents/materials/analysis tools, approved the final draft.
- Xian Wang conceived and designed the experiments, approved the final draft.

## Data Availability

The raw data is available as Data S1.

## Supplemental Information

Supplemental information for this article can be found online at http://dx.doi.org/10.7717/peerj.7252#supplemental-information.

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
