# Peer review of "The prognostic impact of age in different molecular subtypes of breast cancer: a population-based study"

_PeerJ, doi:10.7717/peerj.7252_

## Round 0.1 · original submission · Minor Revisions

Your manuscript has undergone review requires several (minor) issues to be addressed. Please address each issue and provide a point by point response identifying how and where each issue was addressed in the revised manuscript.

Reviewer 1 ·

Basic reporting

The authors attempt at addressing the association between age and breast cancer prognosis is appreciable. The reporting in general is comprehensible and contains appropriate references to existing published studies. Details about the methodology have been included wherever applicable and the data well presented.
I have minor comments with respect to the writing:
Line 47 - the authors fail to mention which proliferative markers they are talking about
Line 48 - the word surrogate to describe the classification of TNBC is not appropriate
Line 54 - ER- has been mentioned twice
Line 58 - was currently controversial should be replaced with subject of discussion or speculation
Line 89 - vital stats instead of vital status

Experimental design

The only, and an important, shortcoming that exists is the fact that age and treatment have not been accounted together and their association has not been evaluated for TNBC deaths for <40 years patients.

Validity of the findings

The authors have addressed the shortcomings of their study, to their full credit. They have also ensured that their interpretations are robust due to the sheer enormity of the sample size taken into consideration along with the large number of variables that have been covered. However, the study falls short on certain minor criteria:
1) No association studies have been performed for patients under 40 who have undergone chemotherapy/ radiotherapy, especially for TNBC samples. This holds important clinical relevance since the authors have concluded how age is an important factor. Their OS survival will thus shed important light in how effective the current therapeutic regimen is.
2) Not much discussion of the results with respect to the clinical implications.
3) I would the authors to speculate (in the discussion section) on the reasons they think TNBC seems to be more prevalent in patients under 40.

Additional comments

Please refer to my specific comments for my minor concerns regarding the study

·

Basic reporting

The manuscript by Dai and Zhong et al. shows the correlation of age with different breast cancer subtypes. The article is too short. Introduction could have been more detailed with more background on different cancer subtypes- for example, which type is more common, aggressive and affects what percent of all breast cancer patients. Table presented is comprehensive though. There are grammatical and typographical errors throughout the manuscript that are stated below in the “Comments for the author” section.

Experimental design

No comment

Validity of the findings

No comment

Additional comments

Some comments, corrections and suggestions are as follows:
1) Line 44: Change “luminal-A and luminal-B” to “luminal A, luminal B”.
2) Line 45: Change “was determined” to “were determined”.
3) Line 54: Change “ER-, ER-” to “ERα, ERβ”.
4) Line 54-55: Replace “higher expression of EGFR” with “higher mRNA expression of HER-2 and EGFR”.
5) Line 57: The suitable word is “older women” instead of “elder women”. This needs to be checked throughout the manuscript.
6) Line 72: Change “United State” to “United States”.
7) Line 74: Change “breast subtypes” to “breast cancer subtypes”.
8) Line 82: Correct by adding underlined word “we involved it with”
9) Line 89: Correct by adding underlined word “should be over”
10) Line 89: Spell-check “recorded”
11) Line 90: Correct by adding underlined word “Patients who did not meet”.
12) Line 104: It looks like something is missing in “the value tumor size” It should be “the values of tumor size.
13) Line 120: Change “was listed” to “is listed”.
14) Line 129: Change “follow-time” to “follow-up time”.
15) Line 137” Did you mean “BCSS” (breast cancer specific survival) instead of “BSCC”. If not, please state what is BSCC.
16) Line 173: Change “deaths rate” to “death rates”.
17) Line 183: Change “cease increasing” to “cease to increase”.
18) Line 189: Correct by adding underlined word “delay in diagnosis”.
19) Line 200: Correct by adding underlined word “found that the”.
20) Line 207: Change “was missed” to “was missing”.

·

Basic reporting

no comment

Experimental design

It should be noticed that which classifaction is used for defining moleculary subgroups clearly in methods.

Validity of the findings

no comment

Additional comments

Dear Authors,
This study is well designed and found out valuable results about breast cancer spesific mortality comparing different range group of age in terms of moleculary subtypes. Figures are well described and have enough quality.
Results are concordant with lieterature; breast cancer patients with TNBC and HR (+), HER 2 (+) moleculary subgroups had higher over all survival in age over 40 years comparing those in age ≤ 40 years. Despite of retrospective design of study patient numbers of study kohort are enough.
It should be noticed that which classifaction is used for defining moleculary subgroups clearly in methods.

---

## Round 0.2 · accepted · Accept

The manuscript has been revised sufficiently and deemed acceptable by reviewers.

Reviewer 1 ·

Basic reporting

No Comments

Experimental design

No comment

Validity of the findings

No Comment

Additional comments

My primary concerns have all been addressed. I would still urge the authors to proof read the manuscript properly to ensure there are no typographical errors. I can still spot a few of them.